# Shade Effect on Phenology, Fruit Yield, and Phenolic Content of Two Wild Blueberry Species in Northwestern Ontario, Canada

**DOI:** 10.3390/plants12244099

**Published:** 2023-12-07

**Authors:** Viktoriya Dyukaryeva, Azim U. Mallik

**Affiliations:** Department of Biology, Lakehead University, Thunder Bay, ON P7B 5E1, Canada; vdyukary@lakeheadu.ca

**Keywords:** blueberry, phenology, reproductive response, vegetative growth, light intensity, black shading net, chemical content, antioxidant capacity, plant physiology

## Abstract

We studied the effect of shade on the phenology, growth, berry yield, and chemical content of two common blueberry species (*Vaccinium myrtilloides* and *V. angustifolium*) in Northwestern Ontario. We hypothesized that high shade would delay vegetative and reproductive phenology and decrease berry yield by increasing resource allocation to vegetative vs. reproductive growth, whereas moderate shade would increase berry phenolic content and antioxidant capacity. We subjected transplanted blueberry plants to a controlled shade treatment and evaluated plant phenological events, vegetative and reproductive growth, berry phenolics, and antioxidant capacity. High shade caused an earlier leaf maturation in *V. myrtilloides,* delayed flowering in *V. angustifolium*, and prolonged fruit maturation in both. The berry yield of both species decreased with increasing shade. High shade reduced the berry phenolic content and antioxidant capacity, especially in *V. myrtilloides*. We concluded that shade shifts species-specific vegetative and reproductive phenology, leading to a difference in resource acquisition, resulting in lower berry yield and antioxidant activity.

## 1. Introduction

Light has a direct effect on photosynthesis, growth, morphological development, metabolism, and reproductive success in plants [1,2]. In combination with other environmental factors, its effect on a plant’s resource metabolism and assimilation varies based on species [3]. Multiple environmental factors, such as soil and air temperature, soil moisture, and fertility, induce morphological and resource allocation plasticity in the *Vaccinium* genus, but very few studies have focused on its response to increasing shade [2,4,5]. Members of the *Vaccinium* genus, including 25 species such as cranberries and blueberries, have phenotypic plasticity that can be attributed to the canopy covers (shade) of a typical boreal forest [4]. Valued for their high nutrient contents and secondary compounds, wild blueberries are an important dietary component for half of 200 forest-reliant communities in Northwestern Ontario [6].

Blueberries, which are rich in sugar, vitamins, amino acids, and enzymes that reduce oxidative cell damage, are well known for their high antioxidant capacity and are often considered a “superfood” [7]. It has been shown that increased UV radiation induces the production of phenolic and anthocyanin compounds in *Vaccinium* species—both strongly associated with antioxidant activity and essential in reversing cellular oxidative damage [8,9]. Oxidative damage is caused to DNA, proteins, and lipids by increased production of reactive oxygen species (ROS) in human cell cultures [10]. Regular dietary intake of one-third of a cup of blueberries, or berry crude extract equivalent, is associated with a reduced risk of cardiovascular disease, death, and type 2 diabetes, as well as improved weight maintenance and neuroprotection [8]. Thus, by investigating the effect of variability of shade-inducing morphological and resource allocation plasticity, the berry yield and antioxidant capacity of *Vaccinium* species have direct ecological, horticultural, and nutritional significance.

Plant phenology studies the timing of initiation and end-of-life cycle events, particularly the developmental stages, annual patterns, biotic and abiotic factors, and their interrelation [11]. Determining a plant’s response to biotic and abiotic factors has wide applications in agriculture, for example, to determine the optimal time for pruning, the application of pesticide and herbicide, and harvest, as well as to characterize carbon balance in terrestrial ecosystems, competition, and the effects of climate change [4,11,12,13,14]. In Northwestern Ontario, the two most abundant blueberry species are velvet leaf blueberry (*V. myrtilloides*) and lowbush blueberry (*V. angustifollium*); the former has been reported to be quite shade tolerant compared with the latter [4,15]. However, no phenological studies have compared the effect of shade on these species.

The boreal forest, the native habitat of Northwestern Ontario blueberries, is characterized by canopy shades that provide a range of light-related microenvironmental conditions [4]. Under such conditions, wild blueberry plants grow vigorously after clearcutting and fire; berry production remains high for 6–8 years and then declines with canopy closure [4]. Some of these factors, such as temperature, precipitation, and humidity, directly affect the development of reproductive structures, where flower development, pollination, fertilization, and fruit maturation are susceptible to microclimatic variation [16,17]. The direct and indirect effects of the microclimate also affect vegetative phenology, plant height, and canopy cover [18]. Overall, the accumulation and longevity of reproductive and vegetative phenological shifts play a vital role in local species distribution and their interaction with neighboring species while also increasing the risk of trophic asynchrony, population declines of higher-level consumers (pollinators and herbivores), and reduction of plant fitness [18,19,20]. 

Photosynthetic processes and carbon assimilation define successful plant performance [21]. Light availability directly affects photosynthesis in leaves and green stems [2]. Under increased shade, plants respond by shifting phenological events and morphological and physiological adjustments to specific leaf area (SLA), internode and petiole lengths, leaf size, leaf thickness, and leaf mass [22]. This allows it to maintain optimal performance under increased shade, a direct plastic response of the active alleviation of environmental stress and reduced resource availability [22]. Shade intensity correlates with the berry yield and quality of blueberry [2]. With prolonged exposure to low light, blueberries experience delayed fruit development and maturation and decreased yield [2]. Moderate shade, either through the artificial use of shading nets or natural canopy overstory, reduces light and temperature stress, which may have a positive effect on the yield and quality of blueberries [2,4]. These favorable environmental factors lead to the increased nutritional value of blueberries and ensure a stable supply of berries for northern communities [4].

The numerous health benefits linked to blueberries come from the array of secondary metabolites, both enzymatic and nonenzymatic, contained in edible berries [7]. In general, genetic background determines the secondary metabolite profile of species, whereas environmental factors, such as light intensity and quality, nutritional status, and water balance, can cause prominent qualitative and quantitative changes to the metabolite composition [23]. In *Vaccinium*, the antioxidant capacity is directly affected by genetic and physiological processes during fruit development and maturation, but their accumulation in stems, leaves, and berries is affected by light and temperature conditions in a species-specific manner [23]. 

The focus of this research was to determine the effects of shade on the phenology, berry yield, and health-related berry chemistry of commonly occurring native Northwestern Ontario blueberry species, *Vaccinium myrtilloides* and *V. angustifolium*. The species were chosen for their major role in local wildlife and human food systems [6,8]. To achieve these objectives, transplanted plants were exposed to a range of experimental shade treatments, and plant phenological events, berry yield, and berry chemistry were recorded.

Our specific objectives were to determine the effect of shade and ground air temperature on (i) phenological events such as vegetative and reproductive growth and (ii) the total phenolic content and antioxidant capacity of blueberries. We hypothesized that (i) deeper shade and lower soil and air temperatures will delay vegetative and reproductive phenological events for both blueberry species, with milder effects in *V. myrtilloides* because of its shade tolerance, (ii) increased shade will decrease berry yield and enhance resource allocation to vegetative organs, resulting in increased leaf area, specific leaf area, and leaf dry matter content in both species, and (iii) moderate shade and milder temperatures will result in the highest secondary metabolite (phenolic) content and, consequently, increased antioxidant capacity. 

## 2. Results

Figure 1 and Figure 2 show the vegetative and reproductive phenologies of *V. angustifolium* and *V. myrtilloides*, respectively. The lines indicate the periods during which the tagged vegetative and reproductive buds were in the indicated phenological stages (Appendix A). Vegetative buds, formed in 2021, were swollen by early May 2022; emerging leaf buds were first observed on 15 May; and 50% of the leaves had unfolded by 25–31 May 2022. At the end of the growing season, more leaves had reached maturity (full leaf unfolding) in both species. With increasing shade, the tagged vegetative buds spent more time in each phenological stage. The leaf development of both species peaked at 80% shade, with 81% of the tagged buds reaching maturity for *V. angustifolium* and 87% reaching maturity for *V. myrtilloides*.

Both blueberry species flowered in late May, and the flowering stage was extended under 80% shade. Immature (green) fruit started emerging 1 week earlier (17 June) in *V. angustifolium* than *V. myrtilloides*, (24 June). For both species, partial shade (30 and 50%) extended the green fruit development period, yielding more fruit than those in full sun and deep shade (Figure 2A,B). Deep shade delayed flowering and produced fewer berries. For example, *V. angustiflium* under 80% shade started flowering on 24 June; in open sun (0% shade), 87% of tagged flower buds produced mature fruits compared with only 33% of buds producing fruits under 80% shade. For *V. myrtilloides*, 86% of tagged flower buds under 0% shade produced mature fruit compared with only 58% of buds under 80% shade.

An inverse relationship was observed between SLA and LDMC in both *Vaccinium* species, in which SLA steadily increased with increasing shade, whereas an opposite trend was observed for LDMC (Figure 3 and Figure 4). Peak values for SLA occurred at 80% shade with 4.57 mm^2^/mg in *V. angustifolium* and 5.48 mm^2^/mg in *V. myrtilloides.* Conversely, peak values for LDMC occurred at 0% shade with 55.71% in *V. angustifolium* and 57.07% in *V. myrtilloides*. Across all shade levels, *V. myrtilloides* had higher SLA (16.85%) and LDMC (9.12%) than *V. angustifolium*.

Both species showed an overall decrease in their reproductive index under increasing shade. *V. angustifolium* had a larger reproductive index than *V. myrtiloides*, peaking at 0 and 50% shade (Figure 5).

The phenolic content of *V. myrtilloides* peaked at 0% shade (594.87 mg), whereas *V. angustifolium* peaked at 30% (310.07 mg), with both peaks falling within the 12–14 °C average seasonal air temperature (Table 1). *V. mytrilloides* berries had higher phenolic content across all shade levels. *V. myrtilloides* had a 59% decrease in total phenolic content under 80% shade compared with its peak under 0% shade (control). *V. angustifolium* showed a 20% decrease between its peak (30% shade) and 80% shade treatment. A similar trend was observed with air temperature, which decreased by 15% from 0% to 80% shade treatments.

The antioxidant capacity of *V. myrtilloides* peaked at 0% shade (107.1 μmol Fe^2+^/g FW), whereas that of *V. angustifolium* peaked at 30% shade (82.4 μmol Fe^2+^/g FW). Both peaks fall within the 12–14 °C average seasonal air temperature (Table 1). *V. mytrilloides* berries had higher antioxidant capacity (FRAP index) across all shade levels. Overall, with increasing shade, both species experienced a decrease in antioxidant activity. The antioxidant capacity in *V. myrtilloides* decreased by 29% from no shade, where it peaked, to 80% shade, whereas *V. angustifolium* experienced a 19% decrease between 30% (its peak) and 80% shade treatments.

## 3. Discussion

This experiment provides novel insight into the effects of shade treatments on the two most common blueberry species in the absence of competition by revealing species-specific phenological responses and fruit chemistry. The long-term implications of the observed vegetative and reproductive phenological delays (Figure 1 and Figure 2) and growth effects can lead to reproductive bottlenecks and an overall decrease in the reproductive fitness of blueberry plants. Delayed flowering mediated decreased reproductive growth and phenological mismatch between blueberries, and their pollinators can have broader negative ecological impacts, resulting in a lower supply of wild blueberries for wildlife species and indigenous peoples in Northwestern Ontario [4].

Our findings regarding the effect of increased shade and decreased air temperature on vegetative and reproductive phenology and growth are consistent with previous results [23]. Increased shade led to more leaf buds reaching maturity across all shade levels, accompanied by larger leaf growth (LDMC and SLA), while *V. myrtilloides* experienced more positive effects across all shade levels. This difference could be attributed to species-specific phenotypic plasticity, with some genetic effect as an adaptation to varied light and temperature conditions enabling it to be more shade tolerant [24]. It is generally believed that the plastic response of SLA enables plants to maintain higher performance under shade, which often results in thinner leaves with lower LDMC [21,25]. This ensures sufficient light capture per gram of leaf tissue and maximizes mass-based photosynthesis [21]. In turn, the delayed flowering observed in both species, with effects generally more prominent in *V. myrtilloides*, can be attributed to differential carbon partitioning by the species, partially driven by their genetic differences [23]. When faced with limiting environmental conditions (low temperature and light), plants employ resource partitioning between reproductive and vegetative meristems, in which a higher vegetative growth (number of leaf buds) tends to delay flowering [23]. It appears that lower soil temperature under increased shade directly led to prolonged fruit maturation in both blueberry species, with *V. myrtilloides* experiencing more dramatic effects. These effects may be indicative of *V. myrtilloides*’ ability to yield berries for a longer period upon canopy closure compared to *V. angustifolium* [23]. Fruit development and maturation are also sensitive to microclimatic conditions. Fruit growth is primarily driven by temperature, which controls cell division and elongation [16]. Increased shading of blueberry plants results in delayed fruit development, and more than 60% shade hinders the photosynthetic ability and growth of leaves [26]. Considering that the timing of reproductive events implies a tradeoff between vegetative and reproductive growth, it is suggested that increased shade also prompts both species to respond to unfavorable conditions by partitioning resources between vegetative and reproductive growth [27]. A similar tradeoff between vegetative and reproductive growth in blueberries grown under 50% shade was reported in a previous study [15].

Both blueberry species had a general decrease in seasonal reproductive growth (fresh berry weight, number, and size) with increasing shade, with *V. myrtilloides* having reduced berry yield across all shade levels (Figure 5). Due to an increase in shade and a decrease in air temperature, late-flowering blueberry plants may have decreased fruit development due to a shortened flowering and fruiting period affecting their reproductive growth [27]. In addition to a shortened available fruit development period, this decrease can be attributed in part to phenological mismatch and pollination time between the blueberry species and their primary pollinators. Blueberry plants require insects, predominantly bees, to cross-pollinate flowers. Greater bee visitation to flowers forms larger fruits that ripen earlier and more evenly [28]. 

The reversal of energy allocation in both species observed in this and our previous experiments can be attributed to both lower temperatures with increasing shade and the absence of interspecies competition. In a common garden experiment, Khan (2015) reported that *V. myrtilloides* produced more berries with increasing shade compared with *V. angustifolium,* a contradictory result of the present study (Figure 5). The enclosed design of the shade structures used in this experiment prohibited pollinators from having easy access to the plants; moreover, shade created a lower air temperature. Considering that flowering was observed earlier in *V.angustifollium* (Figure 1 and Figure 2) and coincided with the peak activity of *Vaccinium*’s primary pollinators, the bees, more of its flowers might have been pollinated than *V. myrtilloides*, leading to a larger yield [28]. These two factors may have contributed to the reversal of the reproductive growth trend observed in Khan’s experiment [15]. 

Members of the *Vaccinium* genus are known for their high phenolic content, which is highly dependent on light intensity, temperature, developmental stage, and species genetic profile [29,30]. In this experiment, both species showed decreased total phenolic content and antioxidant capacity with increasing shade, with *V. myrtilloides* peaking at 0% and *V. angustifolium* peaking at 30%. The literature reports optimal shade levels of 30–50%, with average seasonal temperatures between 12 and 16 ℃ for maximum antioxidant compound production and accumulation in high bush blueberries (*V. corymbosum* L.) [2,31]. Both *V. corymbosum* and *V. angustifolium* are tetraploid with 48 chromosomes, whereas *V. myrtilloides* is a diploid with 24 chromosomes [23,32]. Thus, the decreased phenolic content and antioxidant capacity of *V. angustifolium* under 0% shade could be attributed to a genetic trait of the species (ploidy level).

Mallik and Hamilton (2017) reported that between the two genotypes of blueberries, *V. myrtilloides* has higher total phenolic content [4], as was found in this experiment. The overall higher content of total phenolic and antioxidant activity in *V. myrtilloides* may be a result of a genotypic difference in their ability to synthesize phenolic compounds [33]. This could also be attributed to the difference in the average berry size between the two species. Delayed flowering phenology and decreased pollination may lead to the production of smaller berries in *V. myrtilloides* than in *V. angustifolium* [23]. Smaller berries have a higher average surface area per gram of fresh weight. Considering that antioxidant compounds are largely stored in the berry skin, smaller *V. myrtilloides* berries would yield larger amounts of these compounds [9].

To maximize horticultural yield and health benefits, these blueberry species may be grown under high light and temperate conditions; a coverage of 30% shade is optimal for *V. angustifolium* and 0% shade is optimal for *V. myrtilloides*, both within an air temperature of 12–16 ℃. These results contradict the trends shown by Khan (2014) in their common garden experiment [15]. In the boreal forest, post-fire and post-harvest competition-free time would allow for optimal growth conditions for these species. For the reproductive benefit of the two blueberry species, as well as their primary consumers (forest wildlife and recreational berry pickers), it is imperative that forest fires are not completely suppressed to continue the natural regeneration of blueberries and nutritious food supply. In the absence of fire, forest harvesting using clearcuts can benefit blueberry production. 

This study provides novel insights into the effects of shade and near-ground temperature on Northwestern Ontario blueberry species under controlled shade treatment. The uniform shade provided by the shade structures does not accurately imitate the variable and dynamic shade created by the boreal forest canopy under natural conditions. Furthermore, separation of individual plants removed the effects of inter- and intraspecies competition and possible niche differentiation observed in nature [34]. Future studies should focus on assessing the effects of shade on *V. angustifolium* and *V. myrtilloides* under field conditions [35]. It would be worthwhile to further investigate the phenological responses of blueberries to humidity, pollinators, and the spectral characteristics of light under controlled and field conditions. Measuring photosynthetic activity under different shade treatments would help determine adaptive plasticity traits, such as SLA and LDMC, in relation to variable shade. 

## 4. Materials and Methods

### 4.1. Shade Treatment

Three wooden shade structures were constructed during late April 2022, consisting of a wooden base (244 cm × 244 cm × 152 cm) and a dome-shaped canopy made of PVC pipes. Horticultural shade cloths (Black Greenhouse Shade Cloth 30%–80%, Greek-tek, Inc., Janesville, WI, USA) of select fabric density were used to create 30, 50, and 80% shade treatments. An unshaded space near the shaded domes was used to place control potted blueberry plants. The structures were strategically placed away from buildings or trees on the premises of Lakehead University Greenhouse (Thunder Bay, ON, Canada). The horticultural cloth was placed to avoid overlapping or gaps. 

#### 4.1.1. Blueberry Transplants in Pots

In September 2021, 50 individual ramets (bushes) of *Vaccinium angustifolium* and *V. myrtilloides* were selected based on uniformity of size and transplanted into 5-gallon black plastic pots with 5 cm of native clay-based mineral soil at the bottom and mulched with *Pluerozium schreberi* moss from a 6-year-old clearcut site in the Nipigon area (49°15′38.8″ N 88°22′41.3″ W). The transplanted plants were watered once every 3 days to field capacity until late autumn freezing. 

#### 4.1.2. Phenology and Temperature Logging

In April 2022, 10 blueberry-containing pots of each species were placed under 0, 30, 50, and 80% shade. The ground temperature was measured hourly from 10 May to 28 September 2022, using TEROS 11 + TEROS 12 (METER Group Inc., Pullman, WA, USA) temperature probes on the soil surface connected to Z6L Basic and Em50 data loggers. Soil moisture was maintained at 70% by watering the pots to field capacity every 3 days.

The phenology of each plant was recorded following the method of Fournier et al. (2020) [23]. Before the growing season began, six shoots per plant, three vegetative and three reproductive, were randomly selected and marked with flagging tape. The shoots were selected based on their viability and distance from the surface of the soil (>5 cm). The phenology of all marked shoots was recorded every 3 days from 10 May to 28 September 2022, according to a six-stage development protocol [23]. 

#### 4.1.3. Blueberry Seasonal Growth and Berry Yield

Five randomly selected mature leaves with petioles were collected from each plant on 24 August 2022 and scanned using the HP Smart app at 1200 DPI. Their average leaf area was quantified using WinFOLIA 2022 (Basic version) software (Regent Instruments Inc., Québec, QC, Canada). The fresh and dry weights of the leaves were measured using an analytical scale. The leaves were dried at 35 °C for 72 h using a VWR^®^ Signature™ Forced Air Safety Oven. The leaf dry matter content (LDMC) was calculated as the ratio of leaf dry mass to leaf fresh mass. The specific leaf area was calculated as the ratio of leaf area to leaf dry mass. 

Throughout the season, blueberries from each pot were collected individually as they fully ripened, and their fresh weight was measured. The mean fruit size was determined as a function of the total number of berries per pot divided by their fresh weight. Fruits were stored in a freezer at −18 °C until chemical analysis. The reproductive (berry) yields of both *V. angustifolium* and *V. myrtilloides* during the growing season were expressed as a reproductive index calculated using the berry fresh weight, number, and size per pot. It was created by assigning a score to each of the three parameters. Each point indicated 10 units of increase, either in fresh weight (g), number of berries, or g/berry. The points for the three parameters were summed to yield the “Reproductive Index” score. 

### 4.2. Chemical Analysis

#### 4.2.1. Blueberry Extraction

Solvent extraction was performed using methanol, distilled water, and hydrochloric acid at a ratio of 70:30:1 (organic solvent:water:acid, *v*/*v*/*v*) as the extraction solvent [36]. Blueberry samples (700 mg) were lyophilized and mechanically ground, followed by the addition of 7 mL of extraction solvent. The mixture was homogenized on ice for 4 min [37], then centrifuged at 20,000× *g* for 20 min at 4 °C [9,36]. The supernatant was collected and extraction repeated with the remaining pellet.

#### 4.2.2. Total Phenolic Content

The Singleton and Rossi (1965) methods were used to determine the total phenolic content of blueberry samples [38]. Gallic acid (3,4,5-trihydroxybenzoic acid) was used as the standard. The calibration curve consisted of 100, 250, 500, 750, and 1000 μg/mL Gallic acid solutions. Each standard and blueberry extract (diluted tenfold) were reconstituted with 5.0 mL 2 N Folin Ciocalteu reagent (diluted tenfold). After 5 min, 4.0 mL sodium carbonate solution (75 g/L) was added, and the solution was shaken, closed, and stored in the dark at room temperature for 30 min. Absorbance was measured at 765 nm at 25 °C using a BioTek Synergy H1 reader. 

#### 4.2.3. Ferric Reducing Antioxidant Power Assay (FRAP)

The direct measurement of antioxidant capacity in blueberry samples was performed using the microplate ferric reducing antioxidant power assay (FRAP) described by Bolanos de la Torre et al. (2015) [39]. This FRAP assay was selected based on its ability to provide accurate measurements independent of sample volume effects, as well as because it is inexpensive to perform and easy to calibrate [39]. The calibration curve consisted of 1000, 2000, 3000, 4000, and 5000 μmol/L Fe^2+^ standards prepared daily. DDW was used as instrument zero. For the microplate FRAP assay, sample solution (10 μL), DDW (10 μL), and working FRAP solution (280 μL) were added directly to the 96-well microplate. The plate was shaken and incubated at 37 °C in the dark for 30 min, after which the absorbance was read at 593 nm [39]. 

### 4.3. Data Analysis

Data analysis was performed using a combination of R (version 4.2.0), Paleontological statistics software for analysis and education (version 4.11), and Microsoft Excel (version 2212). Ground temperature readings were averaged in Excel using the ‘=AVERAGE’ function, and daily maximum and minimum temperatures were calculated using ‘=MAX(array)’ and ‘=MIN(array)’ functions, respectively. 

Phenological data analysis was performed manually. For vegetative phenology, the phenological changes were summed every 3 days from 11 May to 18 June 2022. Reproductive phenology changes were summed on a weekly basis from 11 May to 3 September 2022.

Statistical analysis using two-way ANOVA was performed on blueberry fresh weight (g), berry size (g/berry), berry number per pot, leaf area, SLA, and LDCM in the four shade treatments to determine if there was a significant difference in the vegetative growth or yield parameters between *V. myrtilloides* and *V.angustifolium*.

Data on the shade effect on blueberry chemistry (total phenol and antioxidant activity) were analyzed using two-way ANOVA. The differences between the means were evaluated using the Kruskal–Wallis test (*p* < 0.05) and R Studio software version 4.2.0. The results of all determinations are reported as the means ± standard deviations. 

## 5. Conclusions

Increased shade conditions lead to changes in microclimate, as evidenced by the decrease in available light intensity and average air temperature. Together, they contributed to shifts in the reproductive and vegetative phenology of *V. angustifolium* and *V. myrtilloides*, leading to differences in resources, causing decreased berry yield and berry antioxidant properties, with deeper shade producing the largest difference. These plastic responses are a direct consequence of the plant’s effort to maintain optimal activity under decreased light conditions. To maximize horticultural productivity, berry nutritional value, and ecological benefit, *V. myrtlloides* may be grown in 0–30% shade and *V. angustifolium* may be grown in 30–50% shade in a competition-free environment. To fully understand the scope of species-specific phenological responses and adaptive plasticity to changing light and light-related conditions, further investigation should be conducted by simulating field conditions.

## Figures and Tables

**Figure 1 plants-12-04099-f001:**
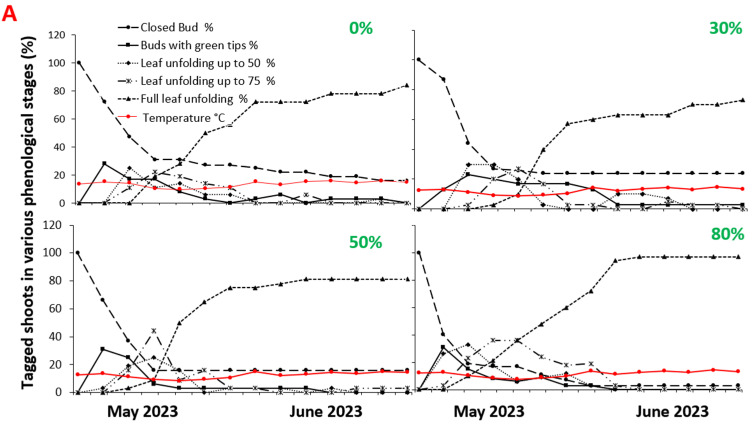
Phenological development of vegetative shoots of: (**A**) *V. angustifolium* and (**B**) *V. myrtilloides* under 0, 30, 50, and 80% shade treatments during the 2022 growing season.

**Figure 2 plants-12-04099-f002:**
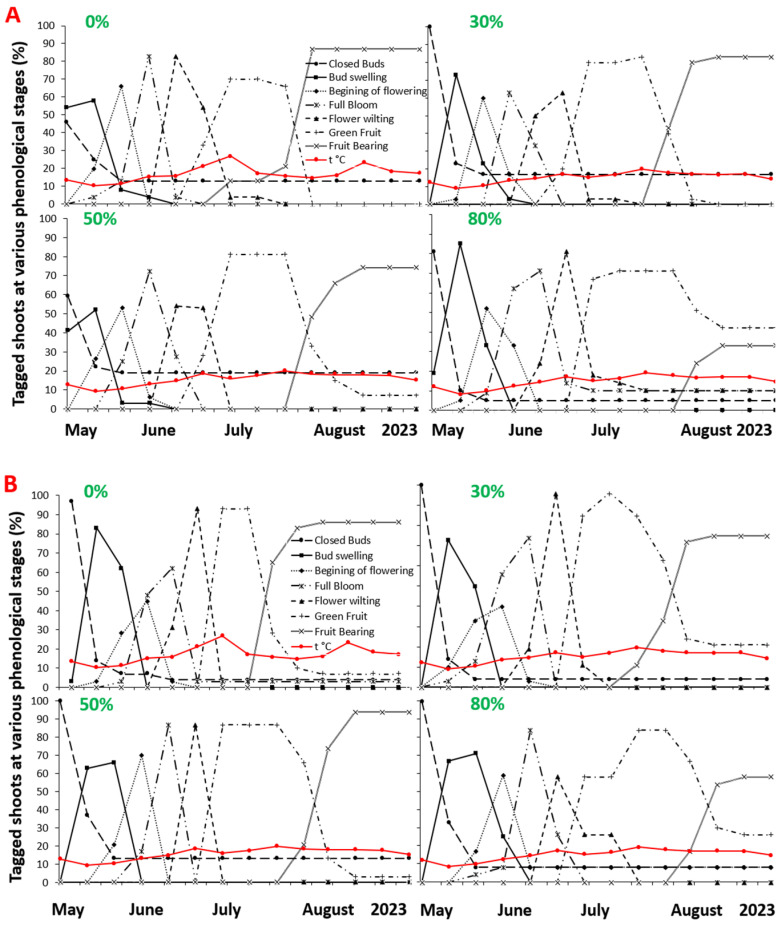
Phenological development of reproductive shoots of: (**A**) *V. angustifolium* and (**B**) *V. myrtilloides* under 0, 30, 50, and 80% shade treatments during the 2022 growing season.

**Figure 3 plants-12-04099-f003:**
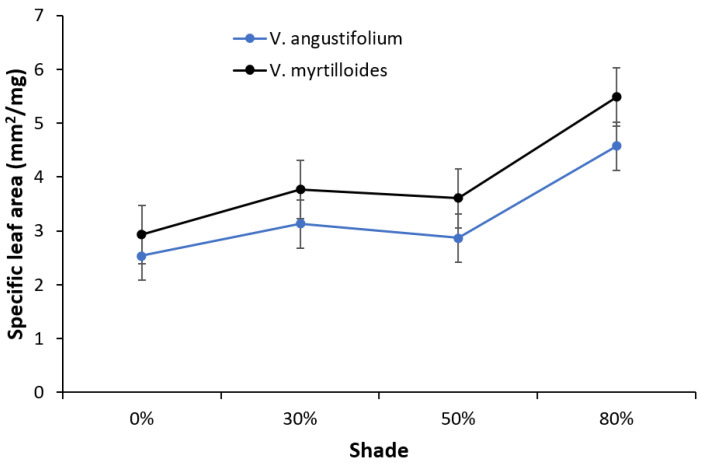
Change in specific leaf area of *V. angustifolium* and *V. myrtilloides* across shade treatments. The error bars represent the standard error in the data.

**Figure 4 plants-12-04099-f004:**
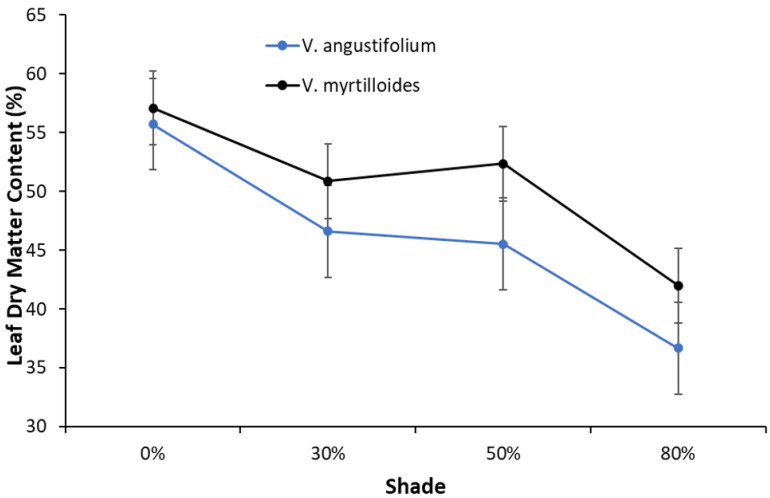
Leaf dry matter contents of *V. angustifolium* and *V. myrtilloides* across shade. The error bars represent the standard error in the data.

**Figure 5 plants-12-04099-f005:**
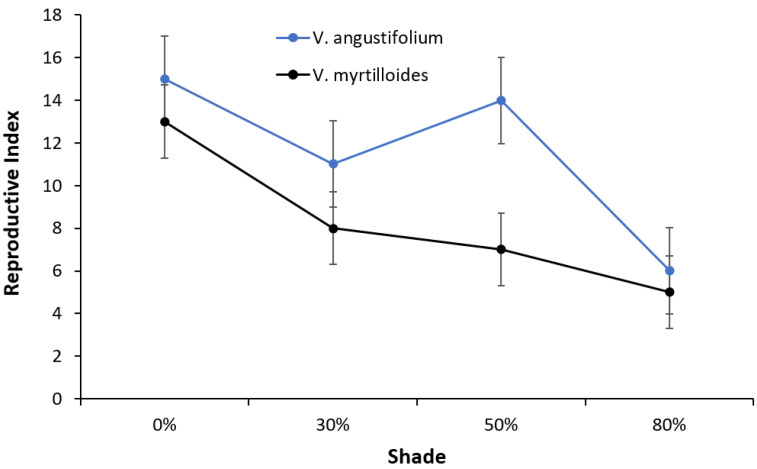
Berry yield index (berry fresh weight, number, and size) of *V. angustifolium* and *V. myrtilloides* across shade treatments. The error bars represent the standard error of mean.

**Table 1 plants-12-04099-t001:** Total phenolic content and antioxidant capacity (as FRAP assay index) of *V. angustifolium* (Va) and *V. myrtilloides* (Vm) with daily average air temperature (*t*) under four shade treatments.

Shade Intensity %	Ground Temp. (°C)	Total Phenol Content (mg GAE/100 g FW)		FRAP Index (μmol Fe^2+^/g FW)	
		Va	Vm	Va	Vm
0	13.5	221.4 ± 20.3 ^a^	594.9 ± 47.2 ^a^	21.1 ± 3.2 ^a,A^	107.1 ± 2.2 ^a^
30	12.4	310.1 ± 29.2	447.2 ± 8.0	82.4 ± 5.5 ^A^	80.8 ± 5.5
50	12.2	305.1 ± 54.8	403.3 ± 94.2	74.5 ± 15.3	75.8 ± 9.2
80	11.5	247.6 ± 36.4	331.9 ± 2.6	67.1 ± 15.7	75.9 ± 0.9

Data are presented as mean ± SD (*n* = 3) per 100 g of fresh weight. The superscripts ^a and A^ denote significant differences (*p* < 0.05) between species and shade treatments, respectively.

## Data Availability

Data is contained within the article and Appendix A.

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
