# Peer review of "Shade Effect on Phenology, Fruit Yield, and Phenolic Content of Two Wild Blueberry Species in Northwestern Ontario, Canada"

_plants, 2023, doi:10.3390/plants12244099_

Round 1
Reviewer 1 Report
Comments and Suggestions for Authors
The submitted manuscript to PLANTS-MDPI entitled “Shade effect on phenology, fruit yield and phenolic content of 2 two wild blueberry species of NW Ontario, Canada” is of great potential to be published. BUT, following are the comments that need to be addressed:
On which basis authors selected these two varieties of blueberry?
Line 8-18: 1-2 sentences are enough for introduction of the study. But authors have given the introduction and M*M, but this information can be shifted to the relevant sections and the main findings, conclusions and future directions should be given in the abstract.
Abstract should be accompanied with the most interesting results.
Line 96: Did authors investigate the effect of temperature as well?
Why did the authors decide to choose FRAP index (μmol Fe2+/g FW) as antioxidant capacity measurement determinant?
It would be even nicer if authors could provide some data about metabolites.
Recommendation: Minor revision

Author Response
The submitted manuscript to PLANTS-MDPI entitled “Shade effect on phenology, fruit yield and phenolic content of 2 two wild blueberry species of NW Ontario, Canada” is of great potential to be published. BUT, following are the comments that need to be addressed:
On which basis authors selected these two varieties of blueberry?
Response: They are the most commonly occurring species in NW Ontario (see ln. 96.)
Line 8-18: 1-2 sentences are enough for introduction of the study. But authors have given the introduction and M*M, but this information can be shifted to the relevant sections and the main findings, conclusions and future directions should be given in the abstract.
Response: Introduction and M & M are revised as per the suggestions.
Abstract should be accompanied with the most interesting results.
Response: Abstract is revised and shortened as suggested (lns. 8-19).
Line 96: Did authors investigate the effect of temperature as well?
Response: Yes, soil surface temperature (see lns. 300-301).
Why did the authors decide to choose FRAP index (μmol Fe2+/g FW) as antioxidant capacity measurement determinant?
It would be even nicer if authors could provide some data about metabolites.
Response: We appreciate the comment. However, this was not the focus of this study. Our objectives were to examine the effect of shade levels on phenological events and health related chemicals (especially antioxidants that protect human cells from ROS mediated damage reported by Sun et al., 2019; ref # 9)
Recommendation: Minor revision
Reviewer 2 Report
Comments and Suggestions for Authors
Generally, it is a well prepared and written study. There are some inconsistencies like a sentence not connected in the Introduction and not complete understandable in the Discussion chapter. Also in Material and methods are missing explications where exactly the plants were placed and from what the substrate consisted. In Conclusions the first sentence is incomplete. In the Reference section the references have to be written correctly: botanical plant names in cursive; the title of the artícles (with some exceptions) should be written in lowercase letters (and not in capital letters). There are some more annotations in the text.

Author Response
Generally, it is a well prepared and written study. There are some inconsistencies like a sentence not connected in the Introduction and not complete understandable in the Discussion chapter. Also in Material and methods are missing explications where exactly the plants were placed and from what the substrate consisted. In Conclusions the first sentence is incomplete. In the Reference section the references have to be written correctly: botanical plant names in cursive; the title of the artícles (with some exceptions) should be written in lowercase letters (and not in capital letters). There are some more annotations in the text.
Response: We appreciate the constructive comments of this reviewer and we thoroughly revised following their recommendations.
Also See the annotated comments on the attached ms pdf
Response: All annotated comments were considered in revising the manuscript.
We thank you and the reviewers for the critical comments and hope that the paper will now be accepted for publication in Plants.
Sincerely,
Azim Mallik